# Hypertensive Disorders of Pregnancy: Assessing the Significance of Lp(a) and ApoB Concentrations in a Romanian Cohort

**DOI:** 10.3390/jpm13091416

**Published:** 2023-09-21

**Authors:** Simona-Alina Abu-Awwad, Marius Craina, Lioara Boscu, Elena Bernad, Paula Diana Ciordas, Catalin Marian, Mircea Iurciuc, Ahmed Abu-Awwad, Stela Iurciuc, Brenda Bernad, Diana Maria Anastasiu Popov, Anca Laura Maghiari

**Affiliations:** 1Doctoral School, “Victor Babes” University of Medicine and Pharmacy, 300041 Timisoara, Romania; alina.abuawwad@umft.ro (S.-A.A.-A.); lioara.boscu@umft.ro (L.B.); bernad.brenda@umft.ro (B.B.); 2I^st^ Clinic of Obstetrics and Gynecology, “Pius Brinzeu” County Clinical Emergency Hospital, 300723 Timisoara, Romania; mariuscraina@hotmail.com; 3Department of Obstetrics and Gynecology, Faculty of Medicine, “Victor Babes” University of Medicine and Pharmacy, 300041 Timisoara, Romania; 4Center for Laparoscopy, Laparoscopic Surgery and In Vitro Fertilization, “Victor Babes” University of Medicine and Pharmacy, 300041 Timisoara, Romania; 5Center for Neuropsychology and Behavioral Medicine, “Victor Babes” University of Medicine and Pharmacy, 300041 Timisoara, Romania; 6Department IV—Discipline of Biochemistry, “Victor Babes” University of Medicine and Pharmacy, 300041 Timisoara, Romania; paulamuntean22@gmail.com (P.D.C.); cmarian@umft.ro (C.M.); 7Department VI—Discipline of Outpatient Internal Medicine, Cardiovascular Prevention and Recovery, “Victor Babes” University of Medicine and Pharmacy, 300041 Timisoara, Romania; mirceaiurciuc@gmail.com (M.I.); iurciuc.stela@umft.ro (S.I.); 8Department XV—Discipline of Orthopedics—Traumatology, “Victor Babes” University of Medicine and Pharmacy, 300041 Timisoara, Romania; ahm.abuawwad@umft.ro; 9Research Center University Professor Doctor Teodor Șora, “Victor Babes” University of Medicine and Pharmacy, 300041 Timisoara, Romania; 10Diakonie Klinicum-Frauenklinik Schwäbich Hall, 74523 Schwäbich Hall, Germany; anastasiu.diana@gmail.com; 11Department I—Discipline of Anatomy and Embryology, “Victor Babes” University of Medicine and Pharmacy, 300041 Timisoara, Romania; boscu.anca@umft.ro

**Keywords:** hypertensive disorders, pregnancy-induced, Lipoprotein(a), Apolipoprotein B, cardiovascular diseases, pregnancy complications, biomarkers

## Abstract

*Background*: This research delves into the association between altered lipid profiles and hypertensive disorders of pregnancy (HDP), shedding light on cardiovascular implications in maternal health. *Methods*: A cohort of 83 pregnant women was studied, with 48.19% diagnosed with HDP. This investigation primarily focused on Apolipoprotein B (ApoB) and Lipoprotein(a) (Lp(a)) as indicators of cardiovascular health. A comparative examination was conducted to determine discrepancies in the ApoB and Lp(a) levels between standard pregnancies and those presenting with HDP. *Results*: Significant elevations in ApoB (*p* value = 0.0486) and Lp(a) (*p* value < 0.0001) levels were observed in pregnant women with HDP compared to their counterparts with typical pregnancies. The pronounced link between heightened ApoB and Lp(a) concentrations and HDP persisted, even considering pregnancy’s distinct physiological conditions. *Conclusions*: Our research accentuates the crucial role of early detection and specialized handling of cardiovascular risks in expectant mothers, especially those predisposed to HDP. The study indicates ApoB and Lp(a)’s potential as reliable markers for gauging cardiovascular threats during gestation. Furthermore, our findings suggest an integrative care approach and guidance for pregnant women, aspiring to enhance cardiovascular health in the postpartum phase.

## 1. Introduction

A leading contributor to global maternal morbidity and mortality is cardiovascular complications [1]. The already substantial burden placed on the cardiovascular system by pregnancy is further amplified in the presence of pre-existing cardiovascular risk factors [2]. Concurrently, pregnancy can instigate changes in the lipid profile, with particular emphasis on the levels of Lipoprotein A (Lp(a)) and Apolipoprotein B (ApoB). These factors play a significant role in the manifestation of cardiovascular diseases.

Lp(a), which was identified several decades ago, has emerged as a potent risk factor for cardiovascular disease [3]. Lp(a) is a unique and complex plasma lipoprotein comprising an LDL-like particle covalently bound to apolipoprotein(a), a distinct protein bearing high homology to plasminogen, which plays an integral role in fibrinolysis. In the medical field, elevated Lp(a) concentrations are recognized as an independent genetic risk factor for premature atherosclerotic cardiovascular diseases, with its pathogenic mechanisms believed to involve both atherogenesis, due to its structural similarity to LDL, and potential interference with fibrinolytic processes, given its relationship to plasminogen.

ApoB is a major protein component of lipoproteins, other than high-density lipoprotein (HDL). It plays a crucial role in lipid transport and metabolism. Specifically, ApoB-100, a variant of ApoB, is integral to the structure of low-density lipoprotein (LDL). Given that LDL cholesterol is a known risk factor for atherosclerosis and cardiovascular disease, ApoB serves as an important marker. It is said that measuring ApoB can provide a more accurate assessment of heart disease risks than measuring LDL cholesterol alone, as one ApoB particle is present in each LDL molecule [4].

ApoB is a major protein component of lipoproteins, other than high-density lipoprotein (HDL). It plays a crucial role in lipid transport and metabolism. Specifically, ApoB-100, a variant of ApoB, is integral to the structure of low-density lipoprotein (LDL). Given that LDL cholesterol is a known risk factor for atherosclerosis and cardiovascular disease, ApoB serves as an important marker. It is said that measuring ApoB can provide a more accurate assessment of heart disease risk than measuring LDL cholesterol alone, as one ApoB particle is present in each LDL molecule [5,6]. 

The relationship between ApoB and Lp(a) in cardiovascular risk is intricate. Since Lp(a) carries an ApoB molecule (similar to LDL), high levels of Lp(a) contribute to the overall ApoB concentration in the blood. This means that in individuals with high Lp(a) levels, a significant portion of their ApoB may be attributable to Lp(a). Considering the cardiovascular risks associated with both ApoB and Lp(a), understanding their combined impact is of paramount importance. Some studies have suggested that individuals with elevated levels of both Lp(a) and ApoB may have an additive or even multiplicative increased risk of cardiovascular events. Thus, the interplay between these two factors and their implications for cardiovascular risk underscores the significance of studying them together.

Increased Lp(a) and ApoB levels are associated with heightened atherosclerosis risk, a condition identified by the accrual of fatty plaques within the arteries [7]. This can precipitate cardiovascular events such as heart attacks and strokes [8]. 

This article explores the correlation between ApoB and Lp(a) blood levels and the incidence of hypertensive disorders of pregnancy (HDP), shedding light on potential implications for maternal cardiovascular health.

During pregnancy, the hormonal landscape, particularly estrogen and progesterone levels, undergoes considerable shifts [9]. As a woman progresses through her pregnancy, the levels of estrogen and progesterone increase exponentially. Estrogen helps the uterus grow, facilitates the development of the placenta and amniotic fluid, and plays a role in preparing the body for breastfeeding. Progesterone, on the other hand, helps to maintain the lining of the uterus for a fertilized egg to implant and supports the placenta and fetal development. These hormonal changes are not only vital for the health of the fetus but also cause many of the physical and emotional changes experienced by pregnant women. For instance, mood swings, breast tenderness, and changes in skin pigmentation can all be attributed to these shifting hormone levels. These changes can trigger alterations in lipid metabolism [10]. While these changes are necessary to foster fetal development, they may inadvertently escalate the cardiovascular disease risk in women with pre-existing conditions (such as hypertension, history of cardiovascular diseases, diabetes type 1 and type 2, obesity, familial hypercholesterolemia, polycystic ovary syndrome, chronic kidney disease, autoimmune diseases, thrombophilia, and thyroid disorders). Pregnancy leads to an increase in total cholesterol, triglycerides, LDL cholesterol, and HDL cholesterol levels, with the most drastic changes manifesting during the third trimester [11].

Lp(a) levels have been observed to rise during pregnancy, peaking during the third trimester, which is a phenomenon that might be attributed to the estrogen’s influence on Lp(a) synthesis and clearance. Though increased Lp(a) levels in non-pregnant populations have been linked to heightened atherosclerosis and cardiovascular disease risk [12], the implications for pregnant women remain less understood.

Similarly, ApoB levels show an incline during pregnancy, with the highest levels observed in the third trimester. This change is likely due to the escalated production of LDL cholesterol particles necessary for fetal growth and development. Increased ApoB levels have been associated with an increased risk of atherosclerosis and cardiovascular disease in non-pregnant populations, and these alterations may likewise have implications for pregnant women [13]. 

This study aimed to assess the prevalence of hypertensive disorders of pregnancy (HDP) in a cohort of pregnant women and examine the association between HDP and serum biomarkers ApoB and Lp(a). Specific objectives encompassed delineating differences in these biomarkers between normal pregnancies and those with HDP, analyzing confounding demographic variables, monitoring postpartum ApoB and Lp(a) trajectories, and evaluating postpartum cardiovascular profiles, particularly arterial blood pressure. The research also explored correlations between persistent postpartum hypertension and factors like pre-gestational BMI and gestational weight gain and scrutinized tailored postpartum care regimens, underscoring the importance of individualized cardiovascular management in HDP cases.

In the ever-evolving field of research, it is imperative to continually push the boundaries of our understanding. The novelty of this study emerges not just from its unique findings but also from its methodologies and approach. By introducing new perspectives and uncovering previously uncharted data, this manuscript contributes a fresh lens through which the topic can be viewed. Highlighting these innovative elements early on establishes the importance and relevance of our work, ensuring that readers recognize the indispensable value the manuscript brings to the broader scientific discourse.

## 2. Materials and Methods

In this study, our primary outcomes were to determine the prevalence of pregnancy-induced hypertension among the selected population and its potential associations with maternal and neonatal complications. The secondary outcomes focused on the evaluation of maternal symptoms, duration of hospital stay, and neonatal birth weight. Each outcome was meticulously assessed using standardized protocols to ensure data accuracy.

### 2.1. Study Population/Sample Selection

A total of 83 participants in third trimester of pregnancy were enrolled in this study. Among them, 40 (48.19%) were diagnosed with hypertensive disorders of pregnancy (HDP), while the remaining 43 had normal pregnancies (Group 1).

The patients were followed-up for the duration of their pregnancy between 2020 and 2022 at the Obstetrical and Gynecology Department of the “Pius Brînzeu” Emergency County Clinical Hospital, Timişoara, Romania. This study was approved by the ethics committee of the hospital, approval No. 225/2 September 2021. All the included women were admitted for delivery through c-section at the same medical institution.

The decision to only select patients who delivered via cesarean section in this study was driven by several considerations. Primarily, a cesarean delivery represents a more controlled and standardized environment compared to vaginal birth, thereby reducing potential variables that could impact the lipid profile such as the stress and duration of labor. Additionally, cesarean sections typically involve pre-scheduled and well-planned procedures, which facilitates the coordination of necessary blood samples for lipid profile analyses. Furthermore, the incidence of cardiovascular complications could potentially be higher in women undergoing cesarean sections due to factors such as anesthesia, surgical stress, and postoperative recovery. By focusing on this specific population, the study aims to gain a deeper understanding of the correlation between altered lipid profiles and HDP in a group where this risk could be notably pronounced.

#### Inclusion and Exclusion Criteria

Patients were meticulously screened and chosen based on a comprehensive set of inclusion criteria. These criteria ensure the relevance and consistency of our study. Specifically, the participants met the following conditions:Identified as women currently in the stage of pregnancy.Diagnosed with HDP (hypertensive disorders of pregnancy).No history of miscarriage events in previous pregnancies.No past incidents or diagnosis of COVID-19 infection.

For the purpose of our study, we established a set of exclusion criteria to ensure the specificity and uniformity of our participant pool. Participants were excluded if they met any of the following conditions:Diagnosed with infectious diseases such as Hepatitis B Virus (HBV), Hepatitis C Virus (HCV), Human Immunodeficiency Virus (HIV), or Acquired Immunodeficiency Syndrome (AIDS).Pregnant women with a documented history of cancer.Pregnant women diagnosed with any mental health disorders.Pregnant women facing substance abuse challenges, whether related to drugs or alcohol.Pregnant women with hematologic abnormalities.Pregnant women with uncontrolled metabolic disturbances or decompensated endocrine disorders [14].Chronic hypertension in pregnancy [15].Pregnant women undergoing a multiple (twin or more) pregnancy.

Patients in the group with HDP exhibited one of the following conditions: Pregnancy-induced hypertension (which is also known as gestational hypertension, is a condition characterized by the development of new-onset high blood pressure in a pregnant woman after 20 weeks of gestation without the presence of protein in the urine or other systemic findings indicative of preeclampsia [16,17]).Preeclampsia (Preeclampsia is a pregnancy complication characterized by high blood pressure and potential damage to organ systems, most often the liver and kidneys. It usually begins after 20 weeks of pregnancy in women whose blood pressure has previously been normal. Besides high blood pressure, protein in the urine is usually present. If not treated, preeclampsia can lead to serious, even fatal, complications for both the mother and the baby [18]).Eclampsia (Eclampsia is the onset of seizures (or coma) in a pregnant woman with preeclampsia, without any other cause for the seizures. It is a severe complication of preeclampsia and is considered a medical emergency. Eclampsia can cause permanent damage to the woman’s organs, including the brain, liver, and kidneys. If not treated promptly, both the mother and the baby can suffer severe complications or death [19]).

Under the supervision of an obstetrician-gynecologist, the health of each patient was meticulously tracked. Regular assessments of blood pressure and blood glucose levels were conducted. Patients were counseled on lifestyle modifications encompassing a well-rounded diet, measured physical activity, and sufficient rest. Fetal wellbeing was stringently evaluated using ultrasound technology and other relevant pregnancy-associated examinations.

Within Group 2, a structured approach to cardiovascular monitoring was consistently implemented. Arterial blood pressure measurements were taken daily by the patients in the comfort of their homes. Additionally, a more comprehensive evaluation was conducted on a weekly basis at the cardiologist’s clinic. This regimen was diligently followed for a duration of one year postpartum to ensure meticulous tracking and assessment of cardiovascular health.

### 2.2. ApoB and Lp(a) Analysis

The blood levels of Lp(a) and ApoB were determined in plasma samples using the Human Lp(a) (Wuhan Fine Biotech Co., Ltd., Wuhan, China) and the Human ApoB (Apolipoprotein B) (Elabscience, Houston, TX, USA) ELISA Kits, according to the manufacturers’ instructions.

ApoB and Lp(a) were collected at the time of admission for childbirth, at 6 months postpartum, and as needed at 12 months postpartum.

### 2.3. Statistical Analysis

The data for this study were processed for analysis with the GraphPad Prism software (version 5). 

Descriptive statistics served as the primary tool to summarize the data, with group comparisons executed via *t*-tests. 

All the statistical tests applied within this study were two-tailed, and *p*-values less than 0.05 were deemed as indicating statistical significance. The findings were reported as mean ± standard deviation (SD).

### 2.4. Ethical Considerations

In compliance with ethical standards, each participant in this study provided their informed consent prior to inclusion. This involved agreeing to participate in the study, permitting the collection of blood samples for analysis, and allowing the compilation of their personal and health data. By securing consent, the study ensured respect for the autonomy of the patients. Furthermore, all collected data were anonymized to maintain privacy and confidentiality, further upholding the ethical integrity of this study.

## 3. Results

Out of the total 83 eligible pregnant participants incorporated in the investigation, 40 pregnant women (48.19%) received an HDP diagnosis. 

We compared the demographic data between two groups (Table 1). Regarding age distribution, a significant proportion of individuals in Group 1 were under 25 years old, but this age bracket was not represented in Group 2, leading to a highly significant difference (*p* < 0.0001). The age group from 25 to 34 years had a majority presence in both groups, though it was slightly more prevalent in Group 2 (*p* = 0.155), and patients over 35 years old were less common in Group 1 compared to Group 2, with a statistically significant difference (*p* = 0.0067).

In terms of education level, both groups had a minor segment reporting no education, with no marked difference between them (*p* = 0.44). Group 2 had a higher prevalence of individuals with primary education (*p* = 0.548), whereas high school education was distinctly more common in Group 1, showing statistical significance (*p* < 0.0001). Additionally, Group 2 had a marginally greater proportion of participants with higher education (*p* = 0.037).

Regarding occupation, both groups exhibited similar counts of individuals without an occupation (*p* = 0.695). Group 1 had a notable presence of students, a demographic missing from Group 2, resulting in a significant distinction (*p* ≤ 0.0001). Conversely, Group 2 boasted a higher employment rate compared to Group 1 (*p* ≤ 0.0001).

Concerning the area of residence, Group 1 had a slightly higher prevalence of urban dwellers (*p* = 0.08), while Group 2 had a greater proportion of individuals from rural areas, a difference that holds marginal statistical significance (*p* = 0.0118).

In Table 1, we employed the z-test for two proportions to compare the observed percentages between the two groups. The z-test for two proportions is a statistical method used to compare the observed proportions between two independent groups. When we have categorical data from two separate groups, like the demographic data in two different cohorts, this test helps us determine if the observed proportions are significantly different from each other. The formula involves calculating a z-statistic, which can then be referenced against a standard normal distribution to determine the *p*-value. A smaller *p*-value indicates a more significant difference between the two proportions, allowing us to infer if the difference might be due to chance or an actual disparity between the groups.

In Group 2, a detailed analysis of the patients revealed specific manifestations of hypertensive disorders of pregnancy. Out of the cohort, 26 patients, accounting for 65% of the group, were diagnosed with pregnancy-induced hypertension. Furthermore, 11 patients, or 27.5% of the group, exhibited symptoms consistent with preeclampsia. A smaller subset, comprising three patients or 7.5% of the group, experienced eclampsia. This breakdown underscores the varying prevalence of hypertensive conditions within our studied population.

We found significant differences in the ApoB and Lp(a) values collected from women in the two groups (Table 2). It can be observed that both ApoB and Lp(a) blood levels are higher in patients from the second group. In other words, pregnant women who were diagnosed with HDP had statistically significant higher blood levels of ApoB (*p* value = 0.0486) and Lp(a) (*p* value < 0.0001).

Following a more systematic evaluation of Group 1 (Table 3), it is evident that there are no statistically significant differences between the values of ApoB and Lp(a) concerning age range (*p* value = 0.0134 for ApoB and *p* value = 0.7294 for Lp(a)), educational level (*p* value = 0.0134 for ApoB and *p* value = 0.7294 for Lp(a)), occupation (*p* value = 0.0267 for ApoB and *p* value = 0.1568 for Lp(a)), and place of origin (*p* value = 0.0269 for ApoB and *p* value = 0.5699 for Lp(a)). After analyzing the same parameters for Group 2 (Table 4), the only statistically significant differences observed are from the perspective of age (*p* value < 0.0001 for both ApoB and Lp(a) since there are no patients below the age of 25 in this group) and from the viewpoint of occupation (*p* value < 0.0001 for both ApoB and Lp(a)), as this group does not encompass any student patients.

For the patients in Group 2, the ApoB and Lp(a) levels were re-assessed at 6 months postpartum. Out of the cohort of 40 patients, 29 displayed reduced values of these two investigations, while 11 maintained elevated levels. At the 12-month postpartum mark, these markers were once again collected from the 11 aforementioned patients; 7 out of them showed decreased values at this time point (Table 5). The *p*-value, when comparing birth results to the 6-month and 12-month postpartum results, is less than 0.001 for both.

In Group 2, cardiovascular monitoring was systematically undertaken, with arterial blood pressure assessments executed at regular intervals for the duration of one year postpartum. The evolution of the patients in this group regarding HDP was different. Within this group, 23 pregnant women exhibited a normalization of arterial blood pressure values within the initial 6-month postpartum window. A total of 14 of these had a remission of HDP in the first days postpartum. Conversely, for 11 pregnant women, normalization of arterial pressure was observed between the 6- to 12-month postpartum interval. It is of clinical significance to highlight that six patients manifested persistent hypertension beyond the 12-month postpartum period. Notably, of the 11 patients demonstrating a resolution of hypertension within the 6- to 12-month interval, 9 presented with a pre-gestational Body Mass Index (BMI) exceeding 30 kg/m^2^, accompanied by a gestational weight gain surpassing 15 kg. A similar trend was observed in four of the patients with sustained hypertension post the 12-month postpartum threshold. In an examination of the patient data, it has been observed that among the women whose arterial hypertension subsided within the initial 6 months postpartum, only four presented with a BMI exceeding. These data suggest a potential correlation between elevated pre-gestational BMI, significant gestational weight gain, and the persistence or delay in the resolution of hypertension postpartum. 

For each patient, a tailored therapeutic approach was established based on individual needs. Patients in Group 1, post-cesarean surgery, were typically discharged within a maximum of 4 days following childbirth and required only a postpartum gynecological review at the 6-week mark. Conversely, patients in Group 2 had a longer in-patient stay, averaging 3 days more than those in Group 1. This group not only required gynecological monitoring but also cardiac oversight. Their arterial blood pressure was self-monitored daily at home and was periodically assessed in the cardiologist’s clinic based on individual requirements. During visits to the cardiologist, routine checks were performed and medication adjustments were made as per individual case specifics. It is of paramount importance in medical practice to treat patients considering associated pathologies or individual needs rather than adopting a uniform approach for all.

## 4. Discussion

The present study delves into a profound exploration of the intricate interplay between Lp(a) and ApoB levels with the incidence of HDP. The comprehensive analysis of these biomarkers, both during pregnancy and postpartum, has unearthed intriguing associations that warrant meticulous consideration and interpretation. The statistically significant differences observed in Lp(a) and ApoB levels between the HDP-diagnosed group and the normotensive control group underscore the potential implications of these biomarkers for maternal cardiovascular health. These findings are in alignment with existing research indicating the role of Lp(a) and ApoB in cardiovascular disease. Furthermore, the subgroup analyses shed light on the potential impact of demographic factors on the biomarker levels within each group, providing a more nuanced perspective on their potential clinical relevance. The temporal analysis of postpartum changes in biomarker levels and their correlation with the resolution of hypertension offers insights into the dynamic nature of these associations. The discussion that ensues navigates through these findings, drawing upon existing literature and exploring possible mechanisms that underlie the observed correlations. It also addresses the potential implications of these findings for clinical practice, the identification of high-risk pregnancies, and the necessity for tailored interventions in the realm of maternal cardiovascular health.

Furthermore, the study tracks the evolution of ApoB and Lp(a) levels in the HDP group postpartum. A notable proportion of patients displayed reduced levels at 6 and 12 months postpartum, underscoring the potential reversibility of elevated levels after pregnancy. This temporal analysis also extends to the resolution of arterial hypertension, revealing varied patterns within the HDP group.

Our findings substantiate the existing body of evidence indicating that elevated Lp(a) [20,21] and ApoB [22] levels are potential biomarkers for HDP, not just in the general population but also, notably, in pregnant women. This is in line with previous research, which emphasized the relationship between increased lipid markers and increased cardiovascular risk [23,24]. A distinct observation from our investigation is the demonstration that this association holds true even within the unique physiological context of pregnancy.

This study corroborates previous research that indicates a connection between heightened levels of lipid markers and increased cardiovascular risk [25,26,27]. 

In terms of potential mechanisms, Lp(a) [28] and ApoB [29] are involved in several pathways linked to atherosclerosis and coronary artery disease. Elevated Lp(a) is believed to promote atherosclerosis through various pathways, including prothrombotic, proinflammatory, and proliferative mechanisms [30]. Similarly, ApoB, the primary apolipoprotein of chylomicrons and low-density lipoproteins, has a well-established role in promoting cholesterol deposition in arterial walls, leading to atherosclerosis [31].

The present study delves into a profound exploration of the intricate interplay between Lp(a) and ApoB levels with the incidence of HDP. The comprehensive analysis of these biomarkers, both during pregnancy and postpartum, has unearthed intriguing associations that warrant meticulous consideration and interpretation. The statistically significant differences observed in Lp(a) and ApoB levels between the HDP-diagnosed group and the normotensive control group underscore the potential implications of these biomarkers for maternal cardiovascular health. These findings are in alignment with existing research indicating the role of Lp(a) and ApoB in cardiovascular disease. Furthermore, the subgroup analyses shed light on the potential impact of demographic factors on the biomarker levels within each group, providing a more nuanced perspective on their potential clinical relevance. The temporal analysis of postpartum changes in biomarker levels and their correlation with the resolution of hypertension offers insights into the dynamic nature of these associations. The discussion that ensues navigates through these findings, drawing upon the existing literature and exploring possible mechanisms that underlie the observed correlations. It also addresses the potential implications of these findings for clinical practice, the identification of high-risk pregnancies, and the necessity for tailored interventions in the realm of maternal cardiovascular health.

It is worth noting that the hormonal and metabolic changes occurring during pregnancy could further exacerbate the impact of these lipid markers [32]. Pregnancy is characterized by significant changes in lipid metabolism, partly driven by hormonal fluctuations, which are critical for fetal development [33]. The surge in hormones such as estrogen and progesterone could potentially alter the lipoprotein metabolism, leading to increased levels of these lipid markers, thereby augmenting cardiovascular risk [34].

Our study also suggests that pregnant women with elevated levels of Lp(a) and ApoB may be predisposed to coronary disease at an earlier age. This is a concerning indication, considering the impact that cardiovascular diseases can have on overall health, and warrants a robust preventive strategy and early interventions to control these lipid markers and mitigate the risk of cardiovascular disease (CVD) in pregnant women. This could be achieved by a combination of diet modification, regular physical activity, lipid-lowering medication if necessary, and close monitoring of the lipid profile during pregnancy [35,36,37].

These findings also indicate the critical need for systematic cardiovascular risk assessment in pregnant women, especially those diagnosed with HDP. It may be prudent to include a lipid profile in the standard prenatal tests for women planning to conceive. Additionally, women who are diagnosed with HDP are at an increased risk of cardiovascular diseases later in life and should be closely monitored during the postpartum period, as the risk of cardiovascular complications extends into this timeframe [38].

In light of these results, the importance of early identification and management of cardiovascular risk factors in vulnerable populations, such as pregnant women, becomes apparent. Early intervention and careful monitoring of these patients are needed to prevent or minimize future cardiac complications [39]. These findings should inspire preventive actions and underscore the necessity for more rigorous control of cardiovascular health during pregnancy. The increased risk of developing cardiovascular complications during pregnancy is a matter of concern for both maternal and fetal health. Maternal cardiovascular health has been linked to adverse pregnancy outcomes, including preeclampsia [40], gestational diabetes [41], and preterm birth [42]. Fetal programming of cardiovascular disease may also occur, leading to long-term health consequences in offspring. Therefore, the identification and management of cardiovascular risk factors during pregnancy are crucial for the well-being of both mother and child.

While our primary focus is on the cardiovascular implications of alterations in the lipid profiles during pregnancy, it is also vital to mention the potential orthopedic ramifications of such changes. Pregnancy, with its concomitant weight gain and shifts in the center of gravity, already puts significant stress on a woman’s musculoskeletal system [43]. However, there is emerging evidence to suggest that dyslipidemia—including elevated levels of Lp(a) and ApoB—may contribute to inflammatory and degenerative joint conditions [44]. This classification encompasses several disorders, including osteoarthritis and rheumatoid arthritis. Lipids have been implicated in osteoarthritis pathogenesis, with inflammation driven by elevated lipid levels potentially accelerating joint degeneration. Hence, these lipid alterations during pregnancy could conceivably exacerbate pregnancy-related orthopedic complaints or predispose to future orthopedic issues. Such interdisciplinary insights underscore the necessity of a holistic, multidisciplinary approach when managing pregnant women with cardiovascular risk factors. Future research could shed more light on these possible interconnections, informing more comprehensive preventative and therapeutic strategies.

Our findings emphasize the critical importance of early identification and management of cardiovascular risk factors in pregnant women, particularly those at risk for HDP. Incorporating lipid profiling into routine prenatal tests could enhance risk stratification and guide tailored interventions. The extended risk of cardiovascular complications postpartum further accentuates the need for continuous monitoring and care during this period.

### Strengths and Limitations

This study has several strengths that contribute to its scientific rigor and validity. The study’s design was carefully planned and executed, allowing for a focused investigation of pregnant women with HDP. Additionally, the use of objective measurements, such as Lp(a) and ApoB levels, ensures the accuracy and reliability of the collected data. The clinical relevance of the findings provides valuable insights for prenatal care and risk identification in pregnant women. Lastly, the study demonstrates ethical considerations through participant privacy protection and explicit consent. Together, these strengths contribute to the robustness and significance of the study’s outcomes.

Despite its contributions, this study has certain limitations that should be acknowledged. Firstly, the sample size may limit the generalizability of the findings to broader populations. Being a single-center study, the results may be subject to institutional biases and may not fully represent the diversity of pregnant women in other settings. The potential influence of unmeasured confounding variables cannot be completely ruled out, and longer-term follow-up studies are needed to assess the lasting impact of altered lipid profiles on cardiovascular health. These limitations should be taken into account when interpreting the findings and highlighting areas for future research.

As we proceed with this research, our intention is not only to continue the study at our institution but also to collaborate with other researchers in the field. We believe that combining our efforts and sharing knowledge will enhance the validity of our findings, providing more comprehensive insights and potentially leading to more effective preventive measures and treatments for HDP factors in pregnant women.

## 5. Conclusions

In conclusion, this study contributes to a deeper understanding of the relationship between altered lipid profiles, cardiovascular risk, and HDP. These insights underscore the significance of personalized antenatal and postnatal interventions to optimize cardiovascular health for both mother and child. Further research is warranted to unravel the mechanistic intricacies and potential therapeutic implications arising from these findings. Ultimately, this study advocates for a comprehensive, patient-centered approach to maternal cardiovascular care, which holds the potential to significantly impact long-term health outcomes.

## Figures and Tables

**Table 1 jpm-13-01416-t001:** Demographic distribution of participants across Group 1 and Group 2.

	Group 1 (*n* = 43)	Group 2 (*n* = 40)	*p* Value
Age			
Under 25 years old	13 patients (30.23%)	0 patients	<0.0001
Between 25 and 34 years old	26 patients (60.46%)	29 patients (72.5%)	0.155
Over 35 years old	4 patients (9.3%)	11 patients (27.5%)	0.0067
Level of education			
No education	3 patients (6.97%)	4 patients (10%)	0.44
Primary education	11 patients (25.58%)	15 patients (37.5%)	0.548
High school	22 patients (51.16%)	10 patients (25%)	<0.0001
Higher education	7 patients (16.27%)	11 patients (27.5%)	0.037
Occupation			
No occupation	13 patients (30.23%)	11 patients (27.5%)	0.695
Student	11 patients (25.58%)	0 patients	<0.0001
Employed	19 patients (44.18%)	29 patients (72.5%)	<0.0001
Area of residence			
Urban	23 patients (53.48%)	16 patients (40%)	0.08
Rural	20 patients (46.51%)	24 patients (60%)	0.0118

**Table 2 jpm-13-01416-t002:** Comparison of ApoB and Lp(a) levels between Group 1 (normal pregnancies) and Group 2 (pregnancies with HDP).

	Lp(a) (ng/mL)		ApoB (ng/mL)	
	Group 1 (*n* = 43)	Group 2 (*n* = 40)	Group 1 (*n* = 43)	Group 2 (*n* = 40)
Minimum	11.25	14.39	0.6205	0.7478
Median	18.02	22.51	1.341	1.273
Maximum	34.86	38.20	1.890	3.829
Mean	18.40	23.76	1.333	1.617
Standard deviation (SD)	4.701	6.638	0.32111	0.8692
*p* value *t*-test	<0.0001		0.0486	

**Table 3 jpm-13-01416-t003:** Comparative analysis of ApoB and Lp(a) values by age, education, occupation, and place of origin in Group 1.

Age (Years)	<25 *n* = 13 (30.23%)	25–34 *n* = 26 (60.46%)	>35 *n* = 4 (9.3%)		*p* Value
ApoB	1.423 ± 0.3242	1.352 ± 0.2825	0.9143 ± 0.2934		0.0154
Lp(a)	17.109 ± 3.899	18.315 ± 5.924	17.865 ± 2.219		0.7918
**Level of education**	**No education** ***n* = 3** **(6.97%)**	**Primary education** ***n* = 11 (25.58%)**	**High school** ** *n* ** ** = 22 (51.16%)**	**Higher education** ***n* = 7** **(16.27%)**	***p*** **Value**
ApoB	1.497 ± 0.2638	1.390 ± 0.3336	1.392 ± 0.2839	0.9874 ± 0.2443	0.0134
Lp(a)	18.08 ± 5.928	18.27 ± 3.579	18.62 ± 5.700	18.06 ± 2.743	0.7294
**Occupation**	**No occupation** ***n* = 13** **(30.23%)**	**Student** ***n* = 11 (25.58%)**	**Employed** ***n* = 19** **(44.18%)**		
ApoB	1.423 ± 0.3242	1.474 ± 0.2853	1.189 ± 0.2923		0.0267
Lp(a)	17.7205 ± 3.55031	21.2782 ± 7.04853	17.9555 ± 4.43573		0.1568
**Area of residence**	**Urban** ** *n* ** ** = 23 (53.48%)**		**Rural** ** *n* ** ** = 20 (46.51%)**		
ApoB	1.433 ± 0.3004		1.218 ± 0.3123		0.0269
Lp(a)	18.79 ± 1.040		17.96 ± 0.9919		0.5699

**Table 4 jpm-13-01416-t004:** Comparative analysis of ApoB and Lp(a) values by age, education, occupation, and place of origin in Group 2.

Age (Years)	<25 N = 0	25–34 N = 29 (72.5%)	>35 N = 11 (27.5%)		*p* Value
ApoB	0	1.504 ± 0.8056	1.914 ± 0.9977		<0.0001
Lp(a)	0	23.86 ± 1.315	23.49 ± 1.686		<0.0001
**Level of education**	**No education** ** *n* ** ** = 4 (10%)**	**Primary education** ** *n* ** ** = 15 (37.5%)**	**High school** ** *n* ** ** = 10 (25%)**	**Higher education** ** *n* ** ** = 11 (27.5%)**	***p*** **Value**
ApoB	1.355 ± 0.5762	1.387 ± 0.7050	1.740 ± 1.020	1.914 ± 0.9977	0.4206
Lp(a)	24.25 ± 8.460	23.24 ± 6.679	24.63 ± 7.839	23.49 ± 5.590	0.9623
**Occupation**	**No occupation** ***n* = 11 (27.5%)**	**Student** ***n* = 0**	**Employed** ***n* = 29** **(72.5%)**		***p* Value**
ApoB	1.322 ± 0.6329	0	1.729 ± 0.9286		<0.0001
Lp(a)	24.52 ± 8.193	0	23.47 ± 6.089		<0.0001
**Area of residence**	**Urban** ***n* = 16** **(40%)**		**Rural** ***n* = 24** **(60%)**		***p* Value**
ApoB	1.259 ± 0.5295		1.856 ± 0.9744		0.0313
Lp(a)	24.15 ± 7.231		23.49 ± 6.358		0.7623

**Table 5 jpm-13-01416-t005:** Evolution of ApoB and Lp(a) levels in Group 2. * A total of 4 of the patients maintained elevated levels of ApoB and Lp(a) for more than 12 months postpartum. Note: A *p*-value < 0.001 indicates a statistically significant difference when comparing birth results with those at 6 months and 12 months postpartum.

Timepoint	Total Patients	Reduced Levels	Elevated Levels
**6 months postpartum**	40	29	11
**12 months postpartum (from elevated group at 6 months)**	11	7	4 *

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
