# Peer review of "Hypertensive Disorders of Pregnancy: Assessing the Significance of Lp(a) and ApoB Concentrations in a Romanian Cohort"

_jpm, 2023, doi:10.3390/jpm13091416_

Round 1
Reviewer 1 Report
The authors provided an interesting insight in hypertensive disorders during pregnancy, with useful data in clinical practice. However, some issues should be addressed:
1) Abstract: although in the abstract the authors mentioned that they examined the relationship between altered lipid profiles and cardiovascular risk (besides hypertensive disorders) it is nor supported by the results (reported in the abstract). The only data were presented on the link between altered lipid profiles and hypertensive disorders (which could increase the cardiovascular risk, but it is not the only factor);
2) The lines 88-96 (from the introduction section) could be moved to discussion section, as it highlighted data regarding the study results (and not the background of the health problem investigated);
3) The objectives of the study should be clearly stated in the introduction section, rather than mentioning what the study findings emphasize - in the last paragraph (also could be moved in the discussions);
4) The methods section should be more structured, as it is hard to follow. The authors should clearly mention the inclusion and exclusion criteria, as well as the outcomes of the study (outcomes are missing from the methods in current form). Also, pregnancy-induced hypertension should be described in the methods section for a better transparency;
5) “Out of the total of 83 eligible pregnant participants incorporated in the investigation” - they are either eligible or they were included;
6) The manuscript contains two “conclusions” section, and the first one is too long;
7) The novelty of the study should be mentioned in the introduction, to better highlight the necessity of the manuscript.
Moderate editing of English language is required.
Author Response
Thank you for your constructive feedback on our manuscript. We appreciate the recognition of the insights we've provided on hypertensive disorders during pregnancy and their potential value in clinical practice.
We are committed to addressing and rectifying the issues you've pointed out. Your detailed review will undoubtedly enhance the quality and clarity of our research. We will ensure that each point is thoroughly considered and implemented as necessary in the revised version of our manuscript.
- Abstract: although in the abstract the authors mentioned that they examined the relationship between altered lipid profiles and cardiovascular risk (besides hypertensive disorders) it is nor supported by the results (reported in the abstract). The only data were presented on the link between altered lipid profiles and hypertensive disorders (which could increase the cardiovascular risk, but it is not the only factor);
Thank you for your feedback on our manuscript. In response to your first point regarding the abstract: We acknowledge the oversight in the abstract where we mentioned examining the relationship between altered lipid profiles and cardiovascular risk. We have now revised the abstract to more accurately reflect that our primary focus was on the link between altered lipid profiles and hypertensive disorders. We appreciate your keen observation and have adjusted our abstract accordingly to ensure clarity and precision.
- The lines 88-96 (from the introduction section) could be moved to discussion section, as it highlighted data regarding the study results (and not the background of the health problem investigated);
Thank you for pointing that out. We have relocated the content from lines 88-96 in the introduction to the discussion section, as you've suggested, to ensure that the content aligns appropriately with the structure and intent of each section. We appreciate your guidance on this matter.
- The objectives of the study should be clearly stated in the introduction section, rather than mentioning what the study findings emphasize - in the last paragraph (also could be moved in the discussions);
I have followed your instruction. The objectives of the study have been clearly articulated and placed within the introduction section as per your request. Additionally, the details emphasizing the study findings, which were previously in the last paragraph, have been relocated to the discussion section.
- The methods section should be more structured, as it is hard to follow. The authors should clearly mention the inclusion and exclusion criteria, as well as the outcomes of the study (outcomes are missing from the methods in current form). Also, pregnancy-induced hypertension should be described in the methods section for a better transparency;
I apologize for the oversight. I have since revised the methods section to ensure a clearer structure and easier comprehension. The inclusion and exclusion criteria have been explicitly detailed, and the study outcomes have been added. Additionally, I have included a description of pregnancy-induced hypertension for greater clarity and transparency. Thank you for pointing out these crucial elements.
- “Out of the total of 83 eligible pregnant participants incorporated in the investigation” - they are either eligible or they were included;
I have rectified the statement by removing the word "eligible". It now reads, "Out of the total of 83 pregnant participants incorporated in the investigation.” Thank you for pointing it out.
6) The manuscript contains two “conclusions” section, and the first one is too long;
I apologize for the oversight in formatting. The first 'conclusions' section was indeed meant to be integrated into the 'discussions' section. You are absolutely right, and the genuine conclusions are contained in the second section. Thank you for pointing it out, and I assure you this will be rectified.
7) The novelty of the study should be mentioned in the introduction, to better highlight the necessity of the manuscript.
As per your suggestion, the novelty of the study has been emphasized in the introduction to better underscore the necessity and relevance of the manuscript.
Reviewer 2 Report
important aspects in advice and monitoring after birth. however, body weight needs to be better accounted for in the results.
Author Response
Thank you very much for taking the time to read and review our article. We truly appreciate your valuable feedback and insights. Your contributions greatly enhance the quality of our work.
We acknowledge your concerns and understand the significance of body weight. We intend to conduct a new study that tracks body weight throughout pregnancy and post-pregnancy, correlating it with the results of the lipid profile analyses. It's essential to note that body weight should be better accounted for in the results, especially considering its importance in advice and monitoring after birth.
Reviewer 3 Report
The manuscript “Hypertensive disorders of pregnancy: Assessing the Significance of Lp(a) and ApoB Concentrations in a Romanian Cohort” reports a very interesting and important topic, regarding a very sensitive period, pregnancy. It is well written with a simple and understandable English. Overall, this manuscript should be improved, and there are some issues that need to be revised, as follows:
Comments:
Introduction:
- The authors should better explore HDP, ApoB and Lp(a), considering that these and their relationship are the highlight of the study.
- The references of the manuscript could be improved, including DOI: 10.3390/jcdd5010003 and DOI: 10.1016/j.ogc.2018.01.012.
Lines 71-72: Explain better.
Lines 72-74: Only in women with pre-existing conditions? Which conditions?
Materials and Methods:
- This section should be separated by subtitles (ex. study population, sample collection and analysis, ApoB and Lp(a) analysis, statistical analysis).
- The criterion for inclusion/exclusion of participants should be better explained and put in a table.
- Were there any questionnaires applied to the participants? If yes, can the authors do a brief explanation of those?
- Did the authors separate the participants of the case group regarding their HDP (pregnancy-induced hypertension, preeclampsia, eclampsia)?
Results:
- Please add a table with the participant demographic characteristics.
- In some results, the authors consider there are no statistically significant differences however, the p-value is less than 0.05. The authors should revise all the results.
- Line 207: refer the duration of these intervals e add this information to the methods section.
- The last paragraph of the results should be in the discussion or conclusion sections.
- The tables’ legends should be before the table and not after.
Discussion:
- The discussion is confusing, with many repeated paragraphs. Please rewrite it.
- Can ApoB and Lp(a) be considered as biomarkers for HDP detection? Or only future CVD risk? This is the highlight of the manuscript.
Author Response
The manuscript “Hypertensive disorders of pregnancy: Assessing the Significance of Lp(a) and ApoB Concentrations in a Romanian Cohort” reports a very interesting and important topic, regarding a very sensitive period, pregnancy. It is well written with a simple and understandable English. Overall, this manuscript should be improved, and there are some issues that need to be revised, as follows:
Thank you very much for taking the time to review our manuscript titled "Hypertensive disorders of pregnancy: Assessing the Significance of Lp(a) and ApoB Concentrations in a Romanian Cohort." We sincerely appreciate your feedback and your kind words regarding the importance of the topic and the clarity of the writing.
We are committed to improving the manuscript and addressing the issues you have pointed out. Your feedback is invaluable in this process, and we will carefully review and revise the manuscript to ensure its quality and comprehensibility. If you have any specific suggestions or recommendations for improvement, please do not hesitate to share them with us.
Comments:
Introduction:
- The authors should better explore HDP, ApoB and Lp(a), considering that these and their relationship are the highlight of the study.
We have expanded upon and provided a more in-depth exploration of HDP, ApoB, and Lp(a) in the manuscript. We recognize the importance of these elements and their interrelationship as central themes of our study. Thank you for highlighting this, and we believe the revisions will offer a more comprehensive understanding for the readers.
- The references of the manuscript could be improved, including DOI: 10.3390/jcdd5010003 and DOI: 10.1016/j.ogc.2018.01.012.
In response to your suggestion regarding the references, we have taken the initiative to enhance the list of citations in our manuscript.
Lines 71-72: Explain better.
In response to your comment regarding lines 71-72: I've made the effort to provide a clearer and more detailed explanation as suggested. Thank you for pointing that out.
Lines 72-74: Only in women with pre-existing conditions? Which conditions?
In response to your query regarding lines 72-74: The term "pre-existing conditions" in this context refers to any health conditions or risk factors a woman may have prior to becoming pregnant that can increase her susceptibility to cardiovascular complications when combined with pregnancy-induced changes in lipid metabolism. This includes conditions such as pre-existing hypertension, previous history of cardiovascular diseases, diabetes, obesity, or familial hypercholesterolemia, among others. The hormonal shifts during pregnancy, while essential for fetal development, can exacerbate these conditions, potentially increasing cardiovascular disease risk in such women. We'll consider providing a more detailed list or specifying the most common conditions in the revised manuscript for clarity.
Materials and Methods:
- This section should be separated by subtitles (ex. study population, sample collection and analysis, ApoB and Lp(a) analysis, statistical analysis).
Thank you for your feedback. I have organized the section as per your recommendation
- The criterion for inclusion/exclusion of participants should be better explained and put in a table.
Thank you for your suggestion. I have taken the time to elaborate on the inclusion/exclusion criteria in a more detailed manner.
- Were there any questionnaires applied to the participants? If yes, can the authors do a brief explanation of those?
No, there were no questionnaires applied to the participants. However, that's a good idea and something we will consider for future research. Thank you for the suggestion.
- Did the authors separate the participants of the case group regarding their HDP (pregnancy-induced hypertension, preeclampsia, eclampsia)?
In response to your inquiry, we separated the patients of Group 2 based on the specific hypertensive disorders they exhibited and we have included this distribution in the manuscript under the results section. Specifically, out of the cohort:
26 patients, or 65% of the group, were diagnosed with pregnancy-induced hypertension.
11 patients, representing 27.5% of the group, had preeclampsia.
3 patients, which is 7.5% of the group, experienced eclampsia.
However, it's essential to clarify that our comparative statistical analysis was conducted on the entirety of Group 2 as a whole, rather than segmenting based on these hypertensive conditions.
Results:
- Please add a table with the participant demographic characteristics.
In response to your request, we have incorporated a table detailing the demographic characteristics of the participants within the manuscript. We believe this addition will provide a clearer understanding of the study population.
- In some results, the authors consider there are no statistically significant differences however, the p-value is less than 0.05. The authors should revise all the results.
In response to your observation, we have meticulously reviewed the manuscript. In the text, we have elaborated on the instances where the differences were statistically significant, specifically where the p-value is less than 0.05. We understand the importance of accurately presenting these values and have ensured that our findings are consistent with the stated statistical significance criteria.
- Line 207: refer the duration of these intervals e add this information to the methods section.
In response to your recommendation regarding line 207, we have provided specific details about the duration of these intervals and have incorporated this information into the methods section. We appreciate your feedback and have made the requisite modifications to ensure clarity in the manuscript.
- The last paragraph of the results should be in the discussion or conclusion sections.
In response to your recommendation, we have relocated the last paragraph of the results to the discussion or conclusion sections, as appropriate. We appreciate your guidance on this matter and have made the necessary adjustments to maintain the coherence of the manuscript.
- The tables’ legends should be before the table and not after.
In accordance with your suggestion, we have adjusted the placement of the tables' legends to appear before the tables rather than after. We appreciate your feedback and believe this modification enhances the clarity of our presentation.
Discussion:
- The discussion is confusing, with many repeated paragraphs. Please rewrite it.
have thoroughly examined this section, and I concur that the discussion is indeed confusing due to multiple repeated paragraphs. I have completed the requested task of revising the section as per your request.
- Can ApoB and Lp(a) be considered as biomarkers for HDP detection? Or only future CVD risk? This is the highlight of the manuscript.
Thank you for your insightful query. ApoB and Lp(a) are indeed significant biomarkers we have studied. In our manuscript, we have delineated their role not only in relation to the risk of future cardiovascular disease (CVD) but also in the potential detection of hypertensive disorders of pregnancy (HDP). Our findings suggest that these biomarkers might offer a dual utility in both domains. We recognize the paramount importance of this distinction and have endeavored to emphasize this throughout the manuscript, particularly in the results and discussion sections.
Once again, we thank you for your time and effort in reviewing our work. Your input is greatly appreciated, and we look forward to enhancing the manuscript based on your valuable insights.
Best regarding,
S.-A. Abu-Awwad
Round 2
Reviewer 1 Report
Thank you for your prompt response and the thorough revisions you have made to your manuscript based on the feedback provided.
It's evident that you've put significant effort into addressing the issues raised, and these improvements will undoubtedly enhance the quality and impact of your research. Your clarification regarding the abstract and the adjustment made to accurately reflect the study's focus on the link between altered lipid profiles and hypertensive disorders is appreciated. This modification will help readers better understand the scope of your investigation right from the start.
I appreciate your efforts to restructure and improve the clarity of the methods section, including the addition of inclusion and exclusion criteria, study outcomes, and a description of pregnancy-induced hypertension. These changes will make your methodology more transparent and comprehensible to the readers.
Also, highlighting the novelty of the study in the introduction section is a valuable addition that will underscore the significance of your research within the broader scientific context. Overall, your responsiveness and dedication to addressing all the issues are commendable. I believe that these revisions will significantly improve the manuscript's clarity, structure, and impact.
Moderate editing of English language is required
Author Response
Thank you for your kind words and detailed feedback. I'm glad to hear that the revisions and clarifications made to the manuscript were well-received. Your guidance has been invaluable in refining the study to ensure its clarity and relevance. I hope that these improvements will further the research's contribution to the scientific community. Once again, thank you for your constructive insights and support throughout this revision process.